# Differences in Selective Attention and Inhibitory Control in Patients with Major Depressive Disorder and Healthy Controls Who Do Not Engage in Sufficient Physical Activity

**DOI:** 10.3390/jcm12103370

**Published:** 2023-05-09

**Authors:** Markus Gerber, Robyn Cody, Johannes Beck, Serge Brand, Lars Donath, Anne Eckert, Oliver Faude, Martin Hatzinger, Christian Imboden, Jan-Niklas Kreppke, Undine E. Lang, Sarah Mans, Thorsten Mikoteit, Anja Oswald, Nina Schweinfurth-Keck, Lukas Zahner, Sebastian Ludyga

**Affiliations:** 1Department for Sport, Exercise and Health, University of Basel, 4052 Basel, Switzerland; 2Psychiatric Clinic Sonnenhalde, 4125 Riehen, Switzerland; 3Adult Psychiatric Clinics (UPKE), University of Basel, 4002 Basel, Switzerland; 4Sleep Disorders Research Center, Kermanshah University of Medical Sciences, Kermanshah 6719851115, Iran; 5Substance Abuse Prevention Research Center, Kermanshah University of Medical Sciences, Kermanshah 6715847141, Iran; 6School of Medicine, Tehran University of Medical Sciences (TUMS), Tehran 1419733141, Iran; 7Department of Intervention Research in Exercise Training, German Sport University Cologne, 50933 Cologne, Germany; 8Psychiatric Services Solothurn, University of Basel, 4503 Solothurn, Switzerland; 9Private Clinic Wyss, 3053 Münchenbuchsee, Switzerland

**Keywords:** Beck Depression Inventory, executive function, flanker task, inhibitory control, major depression, PACINPAT trial, oddball task, symptom severity, sustained attention

## Abstract

Background: Patients with major depressive disorder (MDD) are characterized by neurocognitive impairments and show deficits in various cognitive performance indicators, including executive function. We examined whether sustained attention and inhibitory control differ between patients with MDD and healthy controls, and whether differences exist between patients with mild, moderate, and severe depression. Methods: Clinical in-patients (*N* = 212) aged 18–65 years with a current diagnosis of MDD and 128 healthy controls were recruited. Depression severity was assessed using the Beck Depression Inventory, and sustained attention and inhibitory control were assessed using the oddball and flanker tasks. The use of these tasks promises insights into executive function in depressive patients that are not biased by verbal skills. Group differences were tested via analyses of covariance. Results: Patients with MDD showed slower reaction times in both the oddball and flanker task, independent of the executive demands of the trial types. Younger participants achieved shorter reaction times in both inhibitory control tasks. After correcting for age, education, smoking, BMI, and nationality, only differences in reaction times in the oddball task were statistically significant. In contrast, reaction times were not sensitive to the symptom severity of depression. Conclusion: Our results corroborate deficits in basic information processing and specific impairments in higher-order cognitive processes in MDD patients. As difficulties in executive function underlie problems in planning, initiating, and completing goal-directed activities, they may jeopardize in-patient treatment and contribute to the recurrent nature of depression.

## 1. Introduction

Major depressive disorder (MDD) is one of the most prevalent mental disorders worldwide [1]. In 2017, more than 320 million people were affected [2]. Estimates of lifetime prevalence range between 2 and 30%, dependent on geographic region [3,4,5,6]. During the COVID-19 pandemic, increases in prevalence and disease burden were observed across the world [7]. MDD is associated with high societal costs due to direct health care costs [8], as well as lost work productivity [9]. MDD is often recurrent and relapse is a frequent phenomenon [10]; almost 50% of patients with a first episode of MDD relapse within 2 years after the first episode, and almost 80% relapse during their lifetime [11]. As a consequence, depression often takes a chronic course [12]. MDD contributes to poor quality of life and social adjustment [13,14] and is seen as a leading cause of disability [15]. MDD is highly comorbid with other physical diseases [16] and strongly contributes to the number of years lived with severe impairment [17]. Ultimately, MDD leads to a marked reduction in life expectancy [18].

Nevertheless, physical impairments are not the only way in which depression impacts patients’ lives [19]. Cognitive deficits are also important to understand and explain the impairment of patients with MDD and are likely to be involved in the etiology of the disorder [20]. Depression in mild cognitive impairment is also associated with progression to Alzheimer’s disease [21,22,23]. Difficulties with concentration and decision making have been identified as diagnostic criteria of MDD [24], but deficits seem to extend beyond these core symptoms [25]. Meanwhile, there is solid evidence that patients with MDD are characterized by neurocognitive impairments and show deficits in various cognitive performance indicators, such as attention, memory, problem solving, and processing of emotional information [26,27]. Some scholars have suggested that MDD is associated with a general depletion in cognitive resources [28], whereas others have argued that patients with MDD have sufficient resources but have difficulties with resource allocation and/or the initiation of efficient cognitive strategies [29,30]. Cognitive impairments have been observed in both bipolar [31] and unipolar depression [32].

One form of how cognitive impairments can manifest is in executive function deficits. Executive functions represent a multi-dimensional construct that comprises a set of different (domain-general) cognitive processes relevant for governing goal-directed behavior, including the ability to formulate goals, plan and perform effectively, sustain attention, and regulate emotions [33,34]. Executive functions are especially relevant in response to non-routine situations and are involved in the regulation of various non-executive cognitive processes (e.g., perception, motor responses). Based on the framework provided by Miyake and Friedman [35], inhibitory control is considered a common component of executive function. Inhibitory control encompasses the ability to selectively attend and resist distracting information, thus building on top-down attentional processes [34]. The current body of evidence suggests that selective and sustained attention are both impaired in MDD, which affects behavior in daily life and contributes to a negative attentional bias [36]. The latter describes the tendency of depressed individuals to focus on negative stimuli, and this tendency has a predictive value for the development of depression severity [37]. Executive functions are strongly regulated by prefrontal cortex activity [38]. Thus, executive function deficits have been linked to lesions in the frontal brain regions [39]. These processes are often assessed with laboratory cognitive or neuropsychological tasks [40].

In a recent systematic review, deficits in people with depression were observed in executive subcomponents in 25 of 28 studies [39]. For several reasons, this finding is critical: First, in depressed patients, deficits in executive functions have negative effects on other life domains, including occupational performance, relationships with family members, and life satisfaction [41,42]. Second, empirical evidence suggests that although symptom recovery is paralleled with cognitive improvement, cognitive and executive function deficits can persist even after MDD patients are in remission [43,44]. Third, depressed patients with poor executive dysfunction also show slower and poorer responses to antidepressant treatment [45] and are more vulnerable to relapse [46,47]. Accordingly, executive functions have been proposed as an important target for therapy [25].

In the present study, new data are presented on the relationship between depression and the inhibitory aspects of executive function. The analyses are based on baseline data of a randomized controlled trial which aimed at investigating the effects of a 12-month individually tailored physical activity counseling program on physical activity behavior, fitness, depressive symptom severity, psychological functioning (including anxiety, stress, quality of life, and cognitive function), and cardiovascular risk in a relatively large sample of in-patients with MDD and healthy (non-depressed) controls [48]. As markers of cognitive function, two objective computer-based tests were carried out to assess sustained attention and inhibitory control (oddball and flanker paradigm). The inclusion of these variables was considered important for several reasons. First, previous studies showed that physical activity and cognitive function are interrelated in reciprocal ways [49,50], and executive function can improve as a result of participation in physical activity and exercise training [51,52,53]. However, while these associations are well documented in non-clinical samples, research with psychiatric samples is scarce [54,55,56]. Second, previous studies have shown that self-regulatory control can have an important impact on adherence to physical activity and exercise programs [49,50]. Accordingly, previous studies showed that specific networks and brain areas that are closely involved in inhibitory control are linked to exercise adherence [57,58,59,60]. Therefore, it has been argued that neuropsychological and neurophysiological tests may help predict who is more or less likely to adhere to an intervention, which in turn can help practitioners provide alternative and individualized interventions to individuals with poor expected adherence [61]. However, little is known so far whether such tests have predictive power in psychiatric patients, although it is well known that these patients might experiences difficulties in self-regulatory control [62] and that regular engagement in physical activity is essential to obtain the full benefits associated with a physically active lifestyle [63,64,65]. Third, we decided to focus on the inhibitory aspects of executive function because some authors argued that deficits in executive function in depressed people may be secondary to a primary deficit in attention [66]. This assumption is due to the notion that basic cognitive processes (such as adequate attention) are needed for proper functioning of executive processes. Previous meta-analyses in patients with depression also showed that the strongest deficits appear in tasks that require inhibitory control [67,68].

In the present paper, two research questions were addressed: First, we wanted to find out whether sustained attention and inhibitory control differ in patients with MDD and healthy (non-depressed) controls. Because insufficient physical activity was an inclusion criterion for depressed patients (<150 min of moderate-to-vigorous physical activity per week), healthy controls also had to be insufficiently physically active. Second, we wanted to find out whether sustained attention and inhibitory control differ within our patient sample based on their depressive symptom severity (mild, moderate, severe).

Our study adds to the current literature in several ways: First, existing evidence is based on relatively small samples with typically 20–50 patients [19,39]. Second, few studies have used socio-demographically matched controls to rule out testing effects or other sources of bias [69,70]. Third, different parameters were used to assess executive function in previous research [39]. When measuring inhibitory control, previous investigations have heavily relied on the Stroop Color–Word test [67], despite the criticism that performance on this task is influenced by reading skills [71]. The use of the oddball and flanker tasks promises insights into executive function and attentional processes in depressive patients that are not biased by verbal skills.

Based on the previous literature, three hypotheses were formulated. Firstly (and most generally), we expected poorer performances in sustained attention and inhibitory control in depressed patients compared to healthy controls (Hypothesis 1) [19,39]. Secondly (and more specifically), researchers have suggested two major mechanisms to explain cognitive deficits in MDD, namely the information processing hypothesis and the cognitive effort hypothesis [19,72]. The information processing hypothesis assumes that information processing is slowed down among depressed people, which interferes with higher cognitive function [73,74]. In contrast, the cognitive effort hypothesis suggests that executive dysfunction among depressed patients is not only due to a generalized slowing of information, but rather it also reflects a specific executive deficit. Here, a distinction is made between the execution of automatic tasks that do not require attention and tasks that demand more attentional resources for successful completion [75,76]. As shown in a recent systematic review of 37 studies, both hypotheses are supported by empirical data, showing that compared to healthy controls, both psychomotor slowing and specific executive function deficits are observed in depressed patients [19]. Based on this knowledge, our second hypothesis was that differences will not only appear in indicators that require limited inhibitory control (responses to frequent and congruent stimuli) but also manifest in more non-automatic, effortful tasks (responses to rare and incongruent stimuli) (Hypothesis 2). Thirdly, it has been hypothesized that executive function may vary according to the level of depressive symptom severity [77]. In line with this notion, we expected stronger executive function impairments in sustained attention and inhibitory control among patients with stronger symptom severity [78,79] (Hypothesis 3).

## 2. Materials and Methods

### 2.1. Design

In the present paper, we present baseline data from a multi-centric, two-arm randomized controlled trial (RCT) including an intervention group (extended personalized physical activity and exercise counseling program) and a placebo control group (general instructions about health-enhancing physical activity), allocation concealment, single blinding, and intention-to-treat analysis. The study is a cooperation between four Swiss psychiatric clinics (two public, two private) and the Department of Sport, Exercise and Health of the University of Basel [48].

### 2.2. Participants

Clinical in-patients aged 18–65 years with a current diagnosis of major depression (ICD-10) were recruited between June 2019 and October 2021. For patient screening, a structured clinical interview was carried out by a psychiatrist to ensure that all participants fulfilled the ICD-10 diagnosis for a single episode (F32), recurrent MDD (F33), or bipolar disorder type II (F31-II). The duration of the current depressive episode, number of previous depressive episodes, and psychiatric and somatic comorbidities were also assessed during the process. Symptom severity was assessed using the 17-item Hamilton Depression Rating Scale (HAMD17) [80] by a trained staff member, while patients completed the 21-item Beck Depression Inventory-II (BDI-II) [81]. Level of physical activity during the last week before entering the clinic was measured with the short version of the International Physical Activity Questionnaire (IPAQ) [82]. The inclusion criteria for patients were as follows: (a) women and men, 18–65 years of age; (b) presence of MDD according to the ICD-10 diagnostic criteria; (c) BDI-II ≥ 17 (at least borderline clinical depression); (d) currently not meeting the ACSM physical activity recommendations (IPAQ < 150 min/week of moderate-to-vigorous physical activity); (e) written informed consent; and (f) ability to speak and read German. The exclusion criteria were as follows: (a) presence of history of bipolar disorder type I (F31 type I), history of schizophrenia, or history of schizoaffective disorder; (b) current active alcohol or drug abuse or dependence; (c) any significant medical condition that contraindicates safe participation in physical activity; (d) active suicidal intent; € evidence of significant cardiovascular, neuromuscular, or endocrine disorders limiting regular physical activity, or medical contraindications to physical activity indicated by the Physical Activity Readiness Questionnaire (PAR-Q) [83]; and (f) inability to speak and read German.

To compare baseline differences in the primary and secondary outcomes between patients and healthy controls, we attempted to achieve an age- and gender-matched sample of 167 participants through advertisements in newspapers and through word-of-mouth recommendation. The inclusion criteria for healthy controls were as follows: (a) women and men; (b) 18–65 years of age; (c) HAMD17 ≤ 7; (d) BDI-II ≤ 13; (e) currently not meeting the ACSM physical activity recommendations (IPAQ < 150 min/week of moderate-to-vigorous physical activity); (f) written informed consent; and (g) ability to speak and read German.

In total, we were able to recruit 244 patients and 151 healthy controls who were willing to take part in the study. In total, 212 patients and 128 healthy controls fulfilled all inclusion criteria and had valid data for the oddball and flanker tasks at baseline. The initial power calculation was conducted in light of the expected effect on the primary outcome of the RCT [48]. However, a posteriori power analysis (using G*Power 3.1) showed that our sample was sufficiently powered to detect a small-to-medium effect. Thus, for an ANCOVA with fixed effects, 2 groups, 5 covariates, alpha error = 0.05, and power = 0.80, our sample size allowed us to detect an effect of *f* = 0.153.

### 2.3. Ethical Considerations

Ethical approval was obtained from the “Ethikkommission Nordwest- und Zentralschweiz” (ref approval no. 2018-00976) and all procedures were in line with the ethical principles of the Declaration of Helsinki. The intervention study was registered in the WHO trial register (trial number: ISRCTN10469580). Participants were informed about the general goals of the study, and informed written consent was collected before study entry. Participants were informed that the participation in the study was voluntary and that withdrawal or discontinuation was possible at any time.

### 2.4. Data Assessment and Measures

Screening of patients took place in the first week after admission to in-patient treatment, and baseline data assessment was carried out after 2–3 weeks after admission. Screening and data assessment was combined in the healthy controls and took place at two of the four clinics. Data assessment procedures were identical for the patients and healthy controls.

#### 2.4.1. Depression Severity

Self-reported depression severity was measured using the Beck Depression Inventory-II (BDI-II) [81]. The BDI-II is a 21-item tool, frequently used to assess symptoms of unipolar depression, such as affective, behavioral, cognitive, and somatic symptoms (e.g., “I am so unhappy/sad that I can’t stand it”). Responses were given on a 4-point scale (from 0 to 3). Total BDI-II scores range from 0 to 63, with higher scores reflecting increasing levels of depressive symptomatology. Evidence for the reliability and validity of this measure has been reported previously [84]. Three levels of depression severity were differentiated: mild depression (BDI-II = 0–19); moderate depression (BDI-II = 20–28); and severe depression (BDI-II = 29–63) [85]. Internal consistency of the BDI-II in the present sample was satisfactory (Cronbach’s alpha = 0.94).

#### 2.4.2. Sustained Attention and Inhibitory Control

With regard to cognitive function, we administered computer-based versions of the oddball and flanker tasks with E-Prime 3.0 (PST, USA) to assess inhibitory control. Both tasks are reliable, well-recognized neuropsychological tests for assessing executive function and selective attention in particular [34,86,87,88]. According to the study protocol [48], we also intended to use the 2-back test to assess working memory. However, initial assessments showed that participants had difficulties understanding and focusing on the task. Due to time constraints and to prevent dropouts, we removed the 2-back test from the test battery. This did not affect performance in the other tasks, given that the 2-back test was always applied last in the testing sequence.

The oddball paradigm [89] required participants to press different buttons to deviants (70%) and targets (30%). The letters “X” and “O” were used as visual stimuli and their allocation to the trial type was counterbalanced across the test blocks. Following an inter-trial interval that varied randomly between 800 and 1500 ms, visual stimuli were presented for 250 ms. Responses were collected within 1050 ms following stimulus onset.

For the flanker task [90], participants were instructed to respond to the direction of a centrally presented arrow and to ignore the flanking arrows, which either pointed in the same (congruent trials) or opposite direction (incongruent trials). Both trial types were equiprobable and followed a randomized presentation order. After an inter-trial interval with a random variation between 800 and 1300 ms, visual stimuli were shown for 350 ms, and responses were allowed within 1600 ms. In both the oddball and flanker task, participants completed a practice block and two subsequent test blocks with forty trials each [34,91]. Separately, for each task and trial type, reaction time (on response-correct trials) and accuracy were extracted for statistical analyses. Additionally, the interference score was calculated in the flanker task by subtracting reaction time on congruent trials from the reaction time of incongruent trials. To be included in the data analyses, participants had to achieve accuracy rates, in both tasks, higher than 50%.

#### 2.4.3. Covariates

Participants were asked to report their sex, age, language, nationality, marital status, level of education, employment (rate) prior to hospitalization, years of job experience, smoking status, and whether they had children living at home. Additionally, information about the duration of the current depressive episode, number of prior depressive episodes, age of onset of depression, and current medication was assessed via clinical interview in the patient group.

### 2.5. Statistical Analyses

Information about sample characteristics is presented in form of *M*, *SD*, %, and frequencies. Descriptive statistics (*M*, *SD*) for the different sustained attention and inhibitory control measures are reported separately for the total sample, patients vs. healthy controls, and patients with different depression severity. For the oddball paradigm, the following indicators were considered: accuracy and reaction time for frequent and rare trials. For the flanker task, the indicators were as follows: accuracy and reaction time for congruent and incongruent trials, as well as interference score.

Between-group differences were first tested via univariate analyses of (co)variance with the group (e.g., patients vs. controls) as a fixed factor and the executive function measures as dependent variables. In the case of between-group differences in covariates, these variables were controlled in the ANCOVA. In addition, accuracy was used as a further covariate when reaction times were used as outcomes. If more than two groups were combined, Bonferroni post-hoc tests were applied to further explore which groups differ from each other. In order to consider the notion that accuracy and reaction times in frequent/rare and congruent/incongruent trials might be linked, we further carried out repeated-measures ANCOVA, with the group as a fixed factor, frequent/rare (or congruent/incongruent) as a within-subject factor, and the interaction term between these two factors. All analyses were calculated with SPSS 28 (IBM Corporation, Armonk, NY, USA), and the level of statistical significance was set at *p* < 0.05 across all analyses.

## 3. Results

### 3.1. Sample Characteristics of Patients and Healthy Controls

The characteristics of the total sample, patients with MDD, and healthy controls are shown in Table 1. Sex was equally distributed across groups (54.4% women in the total sample). The mean age was 39.38 years. The majority of the sample spoke German as their first language (87.9%), were Swiss citizens (75.0%), and single (69.4%). Approximately half of the sample reported higher education (47.6%), and one third (31.8%) identified as smokers. Among patients, 34.0 percent had an F32 diagnosis (first episode) and 64.6% had recurrent MDD (F33 diagnosis). At baseline (2–3 after admission to the hospital), 43.4% (*n* = 92) of the patients reported mild symptom severity, whereas 34.0% (*n* = 72) reported moderate depression and 22.6% (*n =* 48) reported severe depression. Moreover, 89.2% of the patients took antidepressants. The healthy controls were younger than the patients, despite our efforts to achieve an age-matched control sample. As shown in Table 1, the patients were more likely to be Swiss, to smoke, and to have higher body weight/BMI, whereas the controls more often reported higher education. The patients also had more years of job experience (mainly because they were older).

### 3.2. Sustained Attention and Inhibitory Control in Patients and Healthy Controls

Table 2 shows that the patients and controls did not differ with regard to accuracy in the two computer-based tasks. Without controlling for covariates, patients differed in reaction time both in response to frequent and rare stimuli in the oddball task. These differences remained after controlling for age, level of education, smoking, BMI, and nationality. The findings showed that patients had longer reaction times for both frequent and rare stimuli. Age was the only significant covariate, showing that younger participants had shorter reaction times (frequent: *p <* 0.001, η^2^ = 0.116; rare: *p* < 0.001, η^2^ = 0.087). After controlling for accuracy and covariates, repeated-measures ANCOVA further showed that reaction times did not differ in response to frequent and rare stimuli (Wilks’ lambda, *F*(1321) = 1.41, *p* = 0.236), that patients had lower reaction times (Wilks’ lambda, *F*(1321) = 6.00, *p* < 0.05, η^2^ = 0.018), and that no interaction existed between type of stimulus and group (Wilks’ lambda, *F*(1321) = 0.244, *p* = 0.622).

In the flanker task, significant differences were found with regard to reaction time in response to congruent and incongruent stimuli. Again, longer reaction times were found in patients compared to controls. In the controlled model, these differences disappeared, mainly because age explained substantial levels of additional variance (congruent: *p* < 0.001, η^2^ = 0.102; incongruent: *p* < 0.001, η^2^ = 0.092), showing that younger participants reacted faster to both types of stimuli. None of the other covariates were associated with the flanker task outcomes. After controlling for accuracy and covariates, the between-group differences reappeared in the repeated-measures ANCOVA, showing that patients had longer reaction times than healthy controls (Wilks’ lambda, *F*(1321) = 4.26, *p* < 0.05, η^2^ = 0.013) but that reaction times in response to frequent and rare stimuli were similar (Wilks’ lambda, *F*(1321) = 0.83, *p* = 0.362), and that there was no interaction between group and type of stimulus (Wilks’ lambda, *F*(1321) = 0.00, *p* = 0.985).

### 3.3. Differences between Patients with Different Symptom Severity

Table 3 shows that patients with mild, moderate, and severe depression did not differ from each other in any of the executive function measures, either with or without the inclusion of covariates. Similarly, in the repeated-measures ANCOVA, no significant main effects were found for type of stimulus or group. Similarly, the interaction terms remained non-significant for both the oddball and flanker task.

## 4. Discussion

The purpose of this paper was to examine whether sustained attention and inhibitory control differ between patients with MDD and healthy controls, and whether differences exist between patients with mild, moderate, and severe depression. Our data show that patients with depression show slower reaction times in both the oddball and flanker task, independent of the executive demands of the trial types. While performance on both tasks differed between groups, reaction times were not sensitive to the symptom severity of depression.

In the present article, we formulated three hypotheses, and each of these are now briefly discussed. Our first and most general hypothesis that patients have poorer performances in executive function than healthy controls was supported. Patients with depression showed slower reaction times on both the oddball and flanker Task (for the flanker task, this result was present only in the uncontrolled analysis), indicating a delay in specific cognitive processes. This effect was not influenced by a speed–accuracy trade-off, given that accuracy was not sensitive to depression. Our findings support previous investigations in which depression was associated with reduced processing speed in various executive function measures [70], deficits in selective attention [92,93], and inhibition [93,94]. These relations have been observed both in the acute phase of the illness and in phases of recovery [76,95,96,97,98]. Possible mechanisms underlying these associations are differences between patients and healthy controls in neurotransmitters, such as dopamine, histamine, or norepinephrine (and associated pathways to the dorsolateral prefrontal cortex), which are important for executive functioning [25]. Prior research has also pointed towards structural and functional abnormalities in prefrontal cortex and connecting subcortical regions, such as the hippocampus, in patients with MDD [99,100]. Because most of the previous studies were based on variants of the Stroop task (e.g., [72,95,97,101,102,103,104]), our findings indicate a generalization of impaired performance to other computer-based tasks that tap into executive function. Knowing that executive function and related attentional processes are essential for building capabilities to make individuals independent, creative, and socially integrated [39,105], impairments in this domain are important targets for intervention. Not surprisingly, therefore, specific types of psychotherapy have been suggested to improve executive function among people with depression, including problem-solving therapy [106], cognitive remediation treatment [107], or cognitive enhancement therapy [108]. It should also be noted that the strength of the differences decreased after inclusion of covariates, which was mainly due to the fact that age explained substantial amounts of additional variance. This is in line with prior investigation showing that in adults, executive function generally declines with increasing age [109,110], an effect that was also observed in people with depression [78,96].

Support was also found for our second hypothesis. Thus, patients with depression showed slower reaction times on trials with lower and higher executive demands. Consequently, depression also manifests in more non-automatic and effortful tasks. This may be due to a general slowing of information processing that also complicates performance on tasks requiring higher-order cognitive functions, which rely on overlapping cognitive processes. Previous findings indicate that information processing and executive function are linked, but that information processing is more sensitive to age [111]. Recalling that controlling for age did not reduce the effect size of depression on trials with low and high executive demands on the oddball task, this argues against the idea that impairments in executive function are due to shared cognitive processes. In contrast, the pattern of findings lends further support for specific deficits in executive function and impairments in selective attention in particular [19].

Our third hypothesis was not supported by our findings as the severity of depressive symptoms did not vary as a function of performance on the oddball and flanker Task. This is at odds with previous studies, which pointed towards lower impairments in subclinical participants compared to clinically depressed patients [78,79]. However, this relation was not reported consistently in previous studies [43,96,104]. This finding also contrasts with previously published results from the PACINPAT study, in which we observed that patients with more severe depression self-report a lower intention to exercise, fewer implantation intentions, more physical activity barriers, and more difficulties in dealing with behavioral obstacles [112]. However, it should be noted that laboratory measures of inhibitory control and self-reported measures of self-control do not necessarily exceed low correlations [94,98,113,114]. While this led some researchers to conclude that computer-based neuropsychological tests have limited value as measures of domain-general inhibitory control [113,115,116], others have recommended the use of both objective and subjective measures, such as the Behavior Rating Inventory of Executive Function—Adult version (BRIEF-A) [117] together to optimize screening [118].

The major strengths of our study are the relatively large sample size (one of the largest samples hitherto assessed to examine the association between depression and executive functions) [19], the use of two tests that have rarely been used in depression research [101,119], and the inclusion of important covariates. Particularly, when comparing patients with mild, moderate, and severe depression, it seemed important to control for depression history because higher impairments were observed in people with recurrent episodes compared to first-episode patients [79,120]. Similarly, we controlled our analyses for antidepressant medication as the impact of medication on cognitive performance is currently a matter of academic debate [121,122]. In line with a recent meta-analysis [122], overall antidepressant treatment was not associated with executive function in the present sample (data not shown). A limitation can be seen in the fact that our results are based on cross-sectional data. Thus, while depression may lead to deficits in executive function via neurophysiological pathways, cognitive vulnerability models of depression [123] have argued that deficits in executive functions may increase the risk of becoming depressed. Thus, attenuated inhibition of negative stimuli in combination with increased activation of subcortical processing regions can contribute to biased attention, distorted cognitions, rumination tendency, and emotional dysregulation caused by the inability to reappraise negative thoughts [124,125,126,127,128]. Additionally, because almost all patients in our sample had an F32 and F33 diagnosis, the findings of our present study cannot be generalized to some other forms of depression. For instance, worse performances in the Stroop task were observed as psychotic compared to non-psychotic depression [129], and poorer performances were found in a meta-analytic study for attention and problem solving in melancholic compared to non-melancholic depression [130]. We also acknowledge that due to the fact that (a) in-patients knew that they would take part in a trial designed to compare two different physical activity counseling approaches, and (b) a key ethical requirement is that participation in scientific studies is voluntary, we cannot fully rule out that in-patients had a more positive attitude towards physical activity than the controls. However, exercise intention was not associated with any of the executive function measures in the present sample (*p* > 0.05). Finally, it can be criticized that we did not use a complete battery of neuropsychological tests in our study, only looking at sustained attention and inhibitory control as two specific types of executive function. However, a meta-analysis yielded similar effect sizes for different executive function tasks in people with depression compared to controls, with *d* = 0.58 for inhibition (*k* = 48); *d* = 0.47 for shifting (*k* = 69); *d* = 0.57 for updating (*k* = 10); *d* = 0.45 for verbal working memory (*k* = 39); *d* = 0.45 for visuospatial working memory (*k* = 23), *d* = 0.38 for planning (*k* = 17); *d* = 0.55 for verbal fluency (*k* = 46); and *d* = 0.33 for processing speed (*k* = 23). Finally, as highlighted in the introduction, inhibitory control seems to reflect general aspects that are common to all other types of executive function [40,98].

From a practical point of view, identification of patients who have reduced sustained attention and inhibitory control, and thus may be more vulnerable to relapse, may facilitate targeted and individually tailored preventive measures to make interventions more effective [98,131]. Without adequate executive function skills, forms of therapy that rely on executive function skills might be less effective [114]. In the realm of our intervention trial, our further analyses will reveal whether patients with higher inhibitory control show better adherence to our lifestyle intervention, whether high adherence mediates the effect of the intervention of physical activity, and whether patients with higher inhibitory control are more likely to show an adequate response to treatment in general.

## 5. Conclusions

Compared to healthy controls, patients with major depression undergoing in-patient treatment show slower reaction time to cognitive tasks that require inhibitory control and sustained attention. This pattern is independent of the executive function demand, suggesting deficits in basic information processing and specific impairments in higher-order cognitive processes. As difficulties in executive function underlie problems in planning, initiating, and completing goal-directed activities, they may jeopardize in-patient treatment and contribute to the recurrent nature of depression.

## Figures and Tables

**Table 1 jcm-12-03370-t001:** Sample characteristics.

	Total Sample (*N* = 340)		Patients with MDD (*n* = 212)		Healthy Controls (*n* = 128)			
	*n*	%	*n*	%	*n*	%	Chi^2^	ϕ
Sex (female)	185	54.4	111	52.4	74	57.8	0.96	0.053
Language (German as first language)	299	87.9	189	89.2	110	85.9	0.78	0.048
Nationality (Swiss)	255	75.0	174	82.1	81	63.3	15.04 ***	0.210
Marital status (single)	236	69.4	150	70.8	86	67.2	0.48	0.038
Level of education (higher education)	162	47.6	76	35.8	86	67.2	31.42 ***	0.304
Children living at home (yes)	77	22.6	26	20.3	51	24.1	0.64	0.043
Antidepressant intake (yes)	189	55.6	189	89.2	0	0.0	253.35	0.869
Smoking (yes)	108	31.8	82	38.7	26	24.1	11.59 ***	0.191
Depression subtype								
Single episode (F32)	---	---	72	34.0	---	---	---	---
Recurrent MDD (F33)	---	---	137	64.6	---	---	---	---
Bipolar disorder type II (F31-II)	---	---	3	1.4	---	---	---	---
	*M*	*SD*	*M*	*SD*	*M*	*SD*	*F*	Eta^2^
Age (in years)	39.38	13.39	41.32	12.83	36.16	13.72	12.23 ***	0.035
Height (in cm)	171.36	9.31	171.64	9.29	170.88	9.35	0.54	0.002
Weight (in kg)	77.19	20.24	80.96	21.24	70.95	16.75	20.65 ***	0.058
Body mass index (kg/m^2^)	26.12	5.98	27.23	6.14	24.29	5.23	20.21 ***	0.057
Employment rate (in hours/week)	24.22	19.76	23.90	20.80	24.74	17.95	0.15	0.000
Years of job experience (in years)	16.79	13.40	18.58	12.86	13.82	13.79	10.34 ***	0.030
Number of children living at home	0.23	0.42	0.24	0.43	0.20	0.40	0.64	0.002
Depressive symptom severity (at baseline)	15.05	11.96	21.69	10.25	4.03	3.36	353.58 ***	0.513
Duration of current episode (in weeks) ^a^	---	---	39.01	51.82	---	---	---	---
Number of prior depressive episodes ^b^	---	---	3.15	6.02	---	---	---	---
Age of onset of depression (in years) ^c^	---	---	29.89	14.14	---	---	---	---

Notes. MDD = Major depressive disorders. ^a^ 32 values missing. ^b^ 17 values missing. ^c^ 16 values missing. *** *p* < 0.001.

**Table 2 jcm-12-03370-t002:** Descriptive statistics and group differences between patients with MDD and healthy controls in executive function measures.

	Total Sample (*N* = 340)		Patients with MDD (*n* = 212)		Healthy Controls (*n* = 128)		Uncontrolled		Controlled for Age, Level of Education, Smoking, BMI, and Nationality	
	*M*	*SD*	*M*	*SD*	*M*	*SD*	*F*	Eta^2^	*F*	Eta^2^
Oddball										
Accuracy: frequent trials	96.2	5.1	95.9	5.3	96.6	4.8	1.42	0.004	0.00	0.000
Accuracy: rare trials	88.2	11.1	87.5	11.1	89.3	11.1	2.3	0.007	0.63	0.002
Reaction time: frequent trials (in ms) ^a^	379.4	68.7	387.16	77.80	366.42	59.22	8.10 **	0.023	6.25 *	0.019
Reaction time: rare trials (in ms) ^b^	410.09	72.64	417.00	76.55	398.63	64.33	8.27 **	0.024	6.09 *	0.018
Flanker										
Accuracy: congruent trials	97.1	5.6	96.7	6.5	97.8	3.5	1.42	0.004	1.55	0.005
Accuracy: incongruent trials	91.4	8.9	90.7	10.1	92.6	0.7	1.96	0.006	1.71	0.005
Reaction time: congruent trials (in ms) ^c^	402.51	77.68	411.44	90.01	387.73	47.88	5.66 *	0.017	1.91	0.006
Reaction time: incongruent trials (in ms) ^d^	445.63	91.93	456.83	108.90	427.06	47.92	6.72 **	0.020	1.86	0.006
Interference score	43.11	41.92	45.39	50.67	39.34	20.07	1.67	0.005	0.23	0.001

Notes. ^a^ Controlled for oddball accuracy for frequent trials. ^b^ Controlled for oddball accuracy in rare trials. ^c^ Controlled for flanker accuracy in congruent trials. ^d^ Controlled for flanker accuracy in incongruent trials. * *p* < 0.05. ** *p* < 0.01.

**Table 3 jcm-12-03370-t003:** Descriptive statistics and group differences between patients with MDD and healthy controls in executive function measures.

	Mild Depression (*n* = 91)		Moderate Depression (*n* = 72)		Severe Depression (*n* = 49)		Uncontrolled		Controlled for Age, Level of Education, Smoking, BMI, Nationality, Depression History (Recurrent/First-Episode), and Antidepressant Medication (Yes/No)	
	*M*	*SD*	*M*	*SD*	*M*	*SD*	*F*	Eta^2^	*F*	Eta^2^
Oddball										
Accuracy: frequent trials	95.6	6.4	96.3	0.4	96.0	4.7	0.40	0.004	0.23	0.002
Accuracy: rare trials	97.3	11.3	88.1	10.8	86.8	11.3	0.23	0.002	0.81	0.002
Reaction time: frequent trials (in ms) ^a^	386.32	60.00	389.63	79.55	385.04	85.15	0.05	0.000	0.86	0.001
Reaction time: rare trials (in ms) ^b^	418.70	70.42	419.91	82.65	409.39	79.43	0.23	0.002	0.91	0.001
Flanker										
Accuracy: congruent trials	96.3	7.4	97.9	3.0	95.6	8.1	2.24	0.021	2.06	0.020
Accuracy: incongruent trials	90.0	11.7	92.7	6.3	89.1	10.7	2.31	0.022	1.83	0.018
Reaction time: congruent trials (in ms) ^c^	416.15	70.17	400.73	64.41	418.47	142.48	0.28	0.003	0.34	0.003
Reaction time: incongruent trials (in ms) ^d^	468.57	114.89	439.79	79.36	459.91	132.09	0.86	0.008	0.59	0.006
Interference score	52.42	70.70	39.06	26.26	41.44	25.09	1.60	0.015	1.05	0.011

Notes. ^a^ Controlled for oddball accuracy for frequent trials. ^b^ Controlled for oddball accuracy in rare trials. ^c^ Controlled for flanker accuracy in congruent trials. ^d^ Controlled for flanker accuracy in incongruent trials.

## Data Availability

Data are available on request due to restrictions.

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
