# Peer review of "Differences in Selective Attention and Inhibitory Control in Patients with Major Depressive Disorder and Healthy Controls Who Do Not Engage in Sufficient Physical Activity"

_jcm, 2023, doi:10.3390/jcm12103370_

Round 1
Reviewer 1 Report
Some comments:
- sample size calculation formula with attention to effect size should be extensively elaborated on.
- attrition and refusal from participants should be reflected on with attention to external validity
- psychometrics of all instruments used should be elaborated on with attention to internal validity.
Author Response
Reviewer 1
Some comments:
Sample size calculation formula with attention to effect size should be extensively elaborated on.
Authors’ response: Thank you for your comment. The a-priori sample size calculation was done for the randomized controlled trial, which was described in the study protocol (Gerber et al., 2019). We have added that an a-posteriori power analysis showed that the sample size was large enough to detect small-to-moderate between-group differences (f=.153) (see lines 240-245).
Attrition and refusal from participants should be reflected on with attention to external validity
Authors’ response: Given the cross-sectional nature of the present data, attrition was not an issue. An inclusion criterion was that all participants need to be insufficiently physically active (£ 150 of moderate-to-vigorous physical activity per week). Due to the fact that (a) in-patients knew that they would take part in a study designed to compare two different approaches towards physical activity counseling and (b) a key ethical requirement is that participation in scientific studies is voluntary, we cannot fully rule out that in-patients had a more positive attitude towards physical activity than controls. However, exercise intention was not associated with any of the executive function measures in the present study (p>.05) (see lines 500-506).
Psychometrics of all instruments used should be elaborated on with attention to internal validity.
Authors’ response: Thank you. Based on your recommendations, we have added Cronbach’s alpha scores for the BDI-II (see lines 271-272). For computerized cognitive tasks, internal validity is not a concern, given that the outcome is not derived from a limited number of items (assessing different concepts), but rather a very high number of trials (assessing the same concept). This is supported by findings indicating that the cognitive processes underlying inhibitory control show high internal consistency after 8-15 trial repetitions (https://doi.org/10.1371/journal.pone.0102672).
Reviewer 2 Report
The manuscript under review focuses on checking the ‘whether sustained attention and inhibitory control differ between patients with MDD and healthy controls, and whether differences exist between patients with mild, moderate and severe depression’.
It should be noted that manuscripts with similar themes using similar methods have already been written, from which it was known that MDD patients respond more slowly than controls. However, the strength of this work is the number of studied indiwiduals at various ages. The research groups were well characterised and the research related to three hypotheses. The discussion, materials and methods and results are clearly written. However, the introduction should be more compact. It is also worth considering, whether to explain in general terms what the Oddball and the Flanker task and what the results are supposed to mean. I think this would make the manuscript easier to understand for less knowledgeable readers.
Author Response
Reviewer 2
The manuscript under review focuses on checking the ‘whether sustained attention and inhibitory control differ between patients with MDD and healthy controls, and whether differences exist between patients with mild, moderate and severe depression’.
It should be noted that manuscripts with similar themes using similar methods have already been written, from which it was known that MDD patients respond more slowly than controls. However, the strength of this work is the number of studied individuals at various ages. The research groups were well characterized and the research related to three hypotheses. The discussion, materials and methods and results are clearly written. However, the introduction should be more compact. It is also worth considering, whether to explain in general terms what the Oddball and the Flanker task and what the results are supposed to mean. I think this would make the manuscript easier to understand for less knowledgeable readers.
Author’s response: Thank you for your positive feedback. In the introduction, we mention that – as markers of cognitive function – two objective computer-based tests were carried out to assess sustained attention and inhibitory control (Oddball and Flanker paradigm) (lines 119-122). As mentioned earlier, sustained attention and inhibitory control are key components of executive function (lines 87-93). In the introduction, we also explain why we focus on these two variables (lines 122-147).
Reviewer 3 Report
In the current paper, the authors explored the differences between patients with major depressive disorder and healthy controls in attention and inhibitory control task performances. Patients with major depressive disorder showed slower reaction time than healthy controls in the user tests.
Overall the manuscript is well written, the design is attractive, and the sample is convenient, the data are well presented, and the discussion of results is accurate. However, I have some minor concerns that the authors should address before publication.
- It seems that the significant difference in age between groups has a not negligible effect on the results, especially regarding the reaction time; could the authors discuss whether a correction for age and educational level (as available for many neuropsychological tests) would have affected the results? A mention of the effect of age and correction by age should be present in the abstract
- the results are limited to attentive and inhibitory control functions; did the authors consider employing a more accurate neuropsychological battery to evaluate executive functions and other cognitive domains? The lack of a complete neuropsychological battery should be added as a limitation.
- I would suggest the authors to briefly discuss in the introduction the link between depression and cognitive impairment and dementia (PMID: 25024328; PMID: 36847011; PMID: 36715000) and the association with subjective cognitive complaints and aging (PMID: 35487700; PMID: 25697700).
- The introduction is extensive and includes information that could be better described in the methods (such as the protocol description, "the analysis are based on baseline data of an RCT..." )
Please define RTC at its first appearance (in the introduction)
Author Response
Reviewer 3
In the current paper, the authors explored the differences between patients with major depressive disorder and healthy controls in attention and inhibitory control task performances. Patients with major depressive disorder showed slower reaction time than healthy controls in the user tests. Overall, the manuscript is well written, the design is attractive, and the sample is convenient, the data are well presented, and the discussion of results is accurate. However, I have some minor concerns that the authors should address before publication.
Author’s response: Thank you for your overall positive feedback. We have addressed all your minor issues in our revised manuscript.
It seems that the significant difference in age between groups has a not negligible effect on the results, especially regarding the reaction time; could the authors discuss whether a correction for age and educational level (as available for many neuropsychological tests) would have affected the results? A mention of the effect of age and correction by age should be present in the abstract.
Author’s response: We agree that the effect of age and correction by age should be mentioned in the abstract (lines 34-37). Since we used age and educational level as covariates, a further correction for these two factors was not needed. Instead, we have added more information of the effect of the considered covariates in the results section (lines 362-364 and lines 375-377).
The results are limited to attentive and inhibitory control functions; did the authors consider employing a more accurate neuropsychological battery to evaluate executive functions and other cognitive domains? The lack of a complete neuropsychological battery should be added as a limitation.
Author’s response: We initially considered to apply one additional test (2-back task) to assess working memory (see study protocol of Gerber et al., 2019). However, after the first few participants, we had to exclude this test because the 2-back task was deemed too taxing by the in-patients. We have added this (see lines 278-283) and the fact that we did not use a complete neuropsychological battery as a limitation in the discussion section (see lines 507-508).
I would suggest the authors to briefly discuss in the introduction the link between depression and cognitive impairment and dementia (PMID: 25024328; PMID: 36847011; PMID: 36715000) and the association with subjective cognitive complaints and aging (PMID: 35487700; PMID: 25697700).
Author’s response: Thank you. We have integrated some of the suggested references, which allowed us to briefly discuss the link between depression and cognitive impairment and dementia (see lines 67-69).